# Exploring Biosensors’ Scientific Production and Research Patterns: A Bibliometric Analysis

**DOI:** 10.3390/s24103082

**Published:** 2024-05-12

**Authors:** Bernardo Valente, Hugo Pinto, Tiago Santos Pereira, Rita Campos

**Affiliations:** 1CES—Centre for Social Studies, Colégio de S. Jerónimo, 3000-995 Coimbra, Portugal; bernardovalente@ces.uc.pt (B.V.); tsp@ces.uc.pt (T.S.P.); ritacampos@ces.uc.pt (R.C.); 2Faculty of Economics & CinTurs—Research Centre for Tourism, Sustainability and Well-Being, University of Algarve, Campus de Gambelas, 8005-139 Faro, Portugal

**Keywords:** biosensors, sustainable MEMS, nanotechnology markets, bibliometric analysis

## Abstract

More sustainable biosensor production is growing in importance, allowing for the development of technological solutions for several industries, such as those in the health, chemical, and food sectors. Tracking the latest advancements in biosensors’ scientific production is fundamental to determining the opportunities for the future of the biosensing field. This article aims to map scientific production in the biosensors field by running a bibliometric analysis of journal articles registered in the Web of Science database under biosensor-related vital concepts. The key concepts were selected by researchers and biosensor technology developers working on the BioAssembler Horizon project. The findings lead to identifying the scientific and technological knowledge base on biosensing devices and tracking the main scientific organisations developing this technology throughout the COVID-19 period (2019–2023). The institutional origin of the publications characterised the global distribution of related knowledge competencies and research partnerships. These results are discussed, shedding light on the scientific, economic, political, and structural factors that contribute to the formation of a scientific knowledge-based focus on the performance and design of these sensors. Moreover, the lack of scientific ties between the three axes of organisations producing expertise in this area (China, USA, and Russia) points towards the need to find synergies through new mechanisms of co-authorship and collaboration.

## 1. Introduction

International institutions have been trying to find new solutions to global ecological issues. The environmental risks, general health degradation, and excessive pollution of urban centres have been addressed in policy-oriented documents such as the EU Green Deal [1] or the UN Sustainable Development Goals [2]. However, investment in Research and Development (R&D) is vital for the further progression of society when finding new technologies to overcome global challenges [3]. 

One of the emerging topics in technology policy scholarships is the influence and impact of biosensors on society. From smartphones to PCR tests, biosensors are crucial for collecting data that can help societies evolve through a digital transition, where it will be easier and more cost-effective to track health conditions or monitor food quality. The production of biosensors is a delicate process that needs to be optimised to achieve a stable and reliable massification of the technology [4], which considers attaining sustainable development goals as one of its fundamental long-term features.

Technology and innovative knowledge sharing have been primarily developed through the growth of unlimited worldwide communication [5]. Following the literature of the field, innovation sharing should be executed within a framework of efficient infrastructural conditions [6], human resources [7] and effective communication channels [8]. Therefore, the different markets of biosensors face two distinct dynamics: a competitive struggle related to the vast number of actors entering the field every day, looking to find room in an ever-growing scientific area, and a need for a cooperation dynamic in R&D to keep upgrading biosensing technologies. 

In this paper, we try to comprehend these divergent dynamics by undertaking a bibliometric analysis of the biosensors scientific field. Through this research, we aim to find the most relevant competencies and organisations for successful innovation in biosensor-related technology. To achieve this, we attempt to track all the influential markets in the biosensors field [9] and look for new actors that are rising through innovative solutions such as multiplex biosensing while using a bibliometric analysis of the most influential research since 2018. 

This work thus looks further into the networks established between “old” and “new” intra- and intercountry actors while drawing forecasting possibilities for the future of the biosensors field. This growth in complexity of the established networks opens new research paths based on understanding the underlying forces behind the relationships between the private and public sectors or between industries and universities/research centres. Aware of this ever-changing dynamic, and by placing it in the centre of the research discussion, this paper aims to provide a map of biosensor technology’s scientific knowledge base and characterise the global distribution of technology and innovation in the field. 

The structure of this paper is built to give appropriate answers to the goals mentioned. It first delves into the biosensors topic by exploring the most relevant literature in the field. After that, the bibliometric analysis methodology is introduced, and the results are displayed through tables and figures based on Web of Science data found appropriate to provide an insight into this field of study. A discussion section follows, assessing the main biosensor scientific clusters regarding the uptake of new competencies required to produce research in the biosensors field; moreover, with modifications in the knowledge base, new research actors are emerging in the area. Finally, the conclusion synthesises this research’s significant findings and bridges the scientific domain with the practical implementation of biosensor innovation by emphasising policy-making recommendations for a sustainable future.

## 2. Exploring the Literature and Applications of Biosensors

Developing innovative and technological products, produced mainly within R&D environments in public and private research centres and enterprises [10], turns a bibliometric analysis into a fundamental tool to detect the most highlighted concerns in the field. Therefore, the technology linkage model has allowed scholars to map the evolution of the technology and nanotechnology field [11]. It also allows the assessment of the gaps in the global biosensors scientific market, which countries’ national authorities can tackle by investing in improving the most valuable competencies. 

From the knowledge acquired by past research on the nanotechnologies topic, there is exponential growth in the literature on the innovation processes established between universities and industries [12]. Nonetheless, some bottlenecks prevent this collaboration and the production of new technologies in an academic environment. Some of the problems concern the lack of flexibility in university procedures and the negotiation process with industries, the obstacles to R&D investment, and the low market potential for patents registered by universities due to the financial or intellectual deficiency of enterprises [13].

In the case of biosensors, the leadership position of Asian scientific production in this field started to be notorious in the last decade. China is still the leading player in that regional cluster; however, in the last century, the remaining BRICS [14] have gained relevance in the field through new STEM networking mechanisms. The USA and the European Union remained the field’s second and third most prolific markets in the same period. However, the gap between the biosensors field in Western countries and that in Asia has increased with the evolution of biosensor gadgets from “proofs-of-concept to device engineering” [15]. 

The biosensors market has seen increasing efforts to produce more mobile health device solutions [16]. These technologies are currently available to the more skilled sectors of the population. Nonetheless, the main goal of portable biosensors is to provide civil society worldwide with democratic biosensing gadgets that allow for the easy reading of outputs and a more accessible screening of results. This effective readability could be complemented by an audio recording or a video of a health professional offering counselling to the patient to deal with the results [17]. 

Focusing on the health-related advantages of biosensors has been the aim of various research studies that also shed light on the trendiest topics in the field’s literature [18,19]. Nanotechnology and biosensors made a significant contribution during the COVID-19 outbreak, when there was an urgency to find suitable technologies for quickly diagnosing the virus and making results readable to all audiences [20]. Diagnosis services previously provided only in hospitals and health facilities penetrated the domestic domain with an associated massification of these biosensing devices. For example, in the USA, the biosensors market grew over 200% during the COVID-19 pandemic years; it is estimated that there will be a compound annual growth rate of 7.5 in the coming years, reaching a total of USD 3.2 billion by the end of 2026 [21]. 

This mass commercialisation and fast adaptation of biosensors to diagnosis gadgets resulted in increased scientific production and partnerships [22,23]. The optimistic approach that most of the literature took on biosensors’ contribution to taming the pandemic also entails new challenges for the future of the technology. However, the ways in which biosensors can evolve the field outside of the framework of PCR technology, which is ubiquitous in COVID-19 diagnosis, are seen as not perfectly accurate and not as cost-effective as other biosensing solutions [24]; some authors considered the pandemic period to be a missed opportunity for the more stable development of different biosensing typologies, such as mechanical biosensors, whose potential was not sufficiently explored during this time [25]. 

Bibliometric analyses of the scientific literature on the diagnostics of Alzheimer’s [26], its application in autoimmune diseases [27], or the prevention of cardiovascular diseases [28] are only some of the examples that emphasise the growth of the biosensors literature in the health sector. This biotechnology focuses on the supply of medical point-of-care solutions, and its usage is also forecasted for other sectors.

It is possible to find studies on the application of this technology in the monitoring and evaluation of wastewater treatment [29], for the detection of pathogens and toxins [30], to ensure an effective agribusiness model [31], and in a broader chemical engineering dimension [32], in which the main discussion is the sensitivity level of biosensor compounds. 

The amount of bibliometric analysis about biosensors in such broad areas of activity sheds light on the indefinite circumstances of this technology. Therefore, several challenges lie ahead, such as the time-consuming activity of building stability in new biosensors [33], the economic performance of the biosensor market in different geographical markets [34], or the ethical issues that arise with the spreading of medical information through biosensors [35]. This grey area of healthcare information can potentially jeopardise the mass employment of big data in healthcare [35]. The interdisciplinarity between the natural sciences and the social sciences and humanities (SSH) can play a decisive role [35]. The SSH must start to be more involved in biosensors’ scientific production to tackle the challenges posed by the interaction between technology development and civil society. 

This interdisciplinarity requires increasing synergies between research centres and scientists from different geographical markets. Channelling international institutions’ willingness to promote sustainability towards the development of biosensor research brings new innovative solutions from diverse cultural backgrounds of science production. Moreover, there is also a diplomatic effort to improve knowledge transfer and production methods between hegemonic poles of technology enhancement, such as the scientific partnership between China and the United States for developing nanotechnology [36]. On the other hand, other literature branches focus on nanotechnology production in markets in Russia’s sphere of influence [37,38,39]. 

To better comprehend the geopolitical, economic, and social framework where the biosensors field is emerging, this work will rely on mapping competencies and technology developers that have contributed the most to the field’s growth. The literature keywords, the number of articles produced, and the geographical location of the most prolific organisations will be determinant features to assess the current state and prospects of the biosensors field.

## 3. Methodology

This study is part of the Horizon Europe project BioAssembler: Integrating bio-inspired assembly into semiconductor manufacturing technology for biosensors, a consortium of partners from Finland, Germany, Austria, and Portugal. The Portuguese partner, the Centre for Social Studies of the University of Coimbra (CES), analyses the socio-economic dimensions of technological innovations in biosensors scientific research and different markets.

The keywords selected for this research were collaboratively chosen by all partners, first by carefully analysing the most recent literature in the biosensors field from different areas of activity and associated markets. The authors participated in several consortium meetings, where the topics and keywords applied to this analysis were discussed and defined. One of the goals of engaging a multiplicity of stakeholders from different backgrounds was to form a holistic framework that would turn this systematic review into useful material for all the sectors of activity related to the production of innovative biosensors. The initial step included delving into the patent database registered at the European Patent Office, and then discussing the most crucial keywords with experts in the biosensors field inside the BioAssembler project. Before the final bibliometric analysis, experimental testing was conducted with Web of Science (WoS) filters using diverse keywords conjugated with Boolean operators that realistically paint a fair portrait of the biosensors field. Several tests were run, with additional keywords inserted, but the results did not change significantly. 

After different stages of topic elimination and steps of fine tuning, a solid agreement between experts and the WoS literature was reached in the keyword definition process, which resulted in the following topics being selected: Photolithograph*; Transducer*; Multiplex Biosensors*; Wafer*; Microchip*; DNA Hybrid*; Synthetic Chemistry*; Silicon Biosensor*; Microelectromechanical Sensor*. The asterisk intends to open the possibilities for words that are not spelt exactly like the defined keywords but are grammatically close to them. The database was quite broad, with various repeated articles, which were erased during the process of fine tuning. Therefore, the selected words passed through several steps to ensure their inclusion would have a meaningful impact on the final result, based on the main article aim, which was to build a landscape based on the literature on biosensor production. The database used for the network analysis contained 77,391 articles, notes, or reviews from the last five years (2018–2022).

These keywords were combined with Boolean operators (“and” and “or”) to understand if clusters or patterns could be extracted from a network analysis of these elements. After successfully determining that most of these topics tended to be aggregated in the literature and utilised simultaneously in the same articles, the literature database was extracted, which served as the input for the VoS Viewer programme. With the help of VoS Viewer, several network maps were drawn in three fundamental dimensions:The first dimension comprises the most common and shared words in the articles extracted through the abovementioned keywords. These are fundamental to understanding the aggregation of competencies and the knowledge base required to develop biosensors.The countries that aggregate themselves depend on the epistemology of the fields and the synergies created between authors from different nations.The organisations that are crucial to comprehend the channels of knowledge production and sharing between universities and research centres.

The network maps and the tables in the following section allowed us to understand the scientific knowledge base of the biosensors industry and to track the most influential actors and how they have partnered to contribute to innovation in the field. This approach, sustained by the employment of a quantitative bibliometric analysis complemented by a qualitative review of the most prolific arguments inside the biosensors field, helped us to evaluate the current state of innovation and elaborate on recommendations for the short-term future of multiplex biosensors.

## 4. Results and Discussion

This section provides a comprehensive overview of the scientific areas gravitating around biosensor technology, which will provide the framework to better understand the field’s knowledge base in the last five years. The competencies that constitute the knowledge base of the biosensors field and the growth of the scientific area are supported by the emergence of new research entities specialised in biosensors. This contributes to an increasing dynamic of universities focused on exploring the ever-growing niche of innovation in biosensors.

### 4.1. Research Areas and Scientific Sources

Table 1 and Table 2 highlight the dispersion of research areas and scientific categories into which biosensors could fit.

From the WoS data, the chemistry field is second to none in the literature related to biosensors. Nonetheless, there is solid versatility in the biosensors field concerning the literature, ranging from engineering to physics or medical subdomains such as optics and oncology. However, based on the bibliometric analysis, the chemistry field is the one that has the most potential to contribute to the scientific expansion of the biosensing devices field. Hence, the following maps (Figure 1 and Figure 2) provide more insight into the competencies that must be further developed to promote a sustainable transition in the chemistry area, and how actors and countries will interact to stimulate new scientific realms and redefine the old ones.

The literature in the biosensors field has been growing steadily. The rise of state-of-the-art technology related to the democratisation of healthcare towards wearable and digital devices turned biosensors and related concepts into a trending topic during the pandemic. The increase in the number of publications in 2020 and 2021 (Figure 1) can be explained by the quick realisation that biosensors could complement the PCR models that began to be produced in massive quantities for COVID-19 diagnosis [20,40,41].

Therefore, the trajectory of the graph in the table follows the findings regarding innovation to tackle the pandemic. In 2022, after the technology for diagnosis had been stabilised and COVID-19’s contagion levels started to decrease, the literature regarding the biosensors topic was reduced to publication numbers close to the ones verified at the beginning of 2019. This was one of the many factors contributing to the decrease in state-of-the-art discussion on the topic in 2022.

The numbers in May 2023 (the month of this article’s data collection) were 3120 articles, notes, or published reviews focused on the words described in the methodology. Scientific production levels are always unpredictable, and they depend on several intricate factors that prevent this work from delving into estimates for 2023. However, if the number of documents produced on the topic in the first five months of 2023 is similar to the rest of the year, there was an even more accentuated fall in the scientific papers published in 2023. As expected, and mentioned above, the journals publishing more biosensor-related research in the last five years were within technology, biology, chemistry, and their subvariants (Figure 2).

The *International Journal of Systematic and Evolutionary Microbiology* and the *Sensors* journal have the largest share of scientific publications in the field. Behind those come journals concerned with molecular science and chemical reactions for developing nanotechnology. The growth of biosensing in science is slowly giving rise to the automatisation of a field of knowledge per se, which can be explained by the ranking of *Biosensors and Bioelectronics* on this list. The future will probably see the more niche journals concerned with only biosensing emerging in this list (cf. Appendix A) simultaneously with steady improvements in sensing technology.

### 4.2. Keywords

The network analysis of the authors’ keywords (Figure 3) shows six agglomerates of knowledge competencies. Firstly, the biosensor group, in green, comprises words such as photolithography, DNA, nanoparticles, and self-assembly. This group occupies a place in the centre of the network and has synergies with all the remaining groups. All five other groups with similar dimensions and relevance are connected to the network’s centre. In yellow/light green, green chemistry and synthetic methods bridge the biosensor group with the agglomerate of liquid chromatography and pharmacokinetics (purple). From these relations of relevance, it is possible to look at the spectrum of knowledge related to green chemistry as an intermediate between the central biosensor-related production group and the possibility of applying it to pharmaceutical and healthcare purposes.

The two groups in blue have a more peripheral relation with the central agglomerate but still emphasise the importance of health-related topics for the field’s literature. Keywords such as hybridisation, stat3, polyphasic taxonomy, and breast cancer are some of the most prominent in these two groups. There is an implicit link between these two groups, more concerned with the functioning of the human body and its components, with the central biosensor group. The largest and most dispersed group is the transducer category (red). The tendency is for this group to establish more connections with other groups as it covers issues in the domain of the utilisation of transducer devices (Figure 4).

On the other hand, the map elaborated based on the WoS keywords has a more homogeneous distribution when compared with the authors’ keywords. In this case, the keywords emphasise the design and level of performance that the sensors might achieve. The vast dimension of new words in the citations’ network analysis, such as expression, derivatives, identification, and chemistry, must be highlighted as it draws new paths to future research towards the chemical analysis of biosensors. For example, the identification concept is crucial within the emerging field of biosensors in the forensic area, which is researching the possibility of identifying criminal details or identities based on biosensors’ detection potential. 

The analysis of the keywords in the database collected in this research points towards the emergence of two areas with particular relevance in the biosensors scientific field. On the one hand, the authors’ keywords are mostly related to chemistry areas dedicated to the development of biosensor technology; conversely, the WoS keywords are primarily related to concerns regarding the application of biosensors and their commercialisation. 

The case of performance and design illustrates the aim of biosensor producers to build more accurate technology with micro-dimensions to be easily operated by the user. Moreover, when looking at the competencies pointed out by the literature, keywords for the development of biosensing imply a concern with sustainability principles. The emphasis on green chemistry and the growth of methods that will allow faster production with little residual waste, such as photolithography or investment in synthetic methods, are symptoms of an R&D field looking for a more environmentally friendly method of market production. 

The WoS keywords complement these advancements as they delve into a different spectrum of analysis regarding the performance and design of the biosensors. This is symptomatic of a convergence in the literature, which further tries to comprehend how changes in the production process can influence the performance, system, and design of devices. There are constant updated guidelines for producing more sustainable and user-friendly devices. The use of these devices to improve medical diagnosis applications is envisioned in the maps, with the word cancer having a significant size in both figures. This mirrors the post-COVID biosensing market tendency of linking the use of this technology to the prevention of large-scale diseases.

### 4.3. Countries

The mapping of the countries’ relationships based on scientific synergies, which mirrors the co-authorship dynamics in the biosensors field, shows new research geographies emerging with the expansion of the concepts being addressed by biosensors R&D. Innovation has gravitated around publications on biosensors in recent years, including the growth of social sciences and humanities approaches towards technology development, which triggers the appearance of countries coming from an academic tradition of critical science and technology studies. 

An evaluation of countries by the number of publications (Figure 5) portrays the People’s Republic of China (orange) as having the most significant number of publications. However, the small number of links with other scientific institutions outside of Asia pushes China’s hegemonic position to the outskirts of the map. On the other hand, many countries outside the most well-known geographies for biosensor production have registered substantial partnership synergies, which mean those countries are closer to the centre of the network. The examples of Germany (red), France (green), and England’s (light blue) relevant positions are supported by the strong connections between anglophone and European research traditions of knowledge transfer. The USA is the core of this sub-group, emphasising its intermediary position that bridges biosensor-related scientific production in European research centres with more peripheral geographies.

Spread throughout the map are some countries that, despite having numerous records of publications, occupy less relevant positions, which is the case of Brazil and Spain (yellow/light green), with strong connections to the South American market but a weak link with more central countries. This cluster is an example of a growing broad scholarship on science and technology studies (STS) and culture, emphasising the role of sharing as a crucial feature of knowledge transfer in the field [42]. 

The hierarchy of scientific relationships in the biosensors field within the Middle East has a similar structure of epistemic community to the ones formed by Southeastern Asian nations. In the case of the Middle East, the group’s core lies in Iran, and there are prominent relationships with other Maghreb nations; however, similar to the Southeast Asian example, it misses scientific partnerships with R&D entities outside the Middle East/Northern Africa. Therefore, the diverse sub-groups of knowledge transfer in Asia within the field of biosensor technology, in which the links are heavily based on countries’ geographic proximity, must develop new collaboration mechanisms with countries outside their epistemic communities. 

When considering the network distribution of countries by the number of citations (cf. Appendix A), all the countries gravitate around the centre of the map, establishing strong links between each other. The People’s Republic of China is not only the country with the highest scientific production in the biosensors field but is also the most cited country. On the other hand, the USA occupies a secondary role concerning scientific citations. However, it leads consortiums with a pack of R&D institutions in the anglophone world, allowing the USA to create interconnections with all the other medium-sized countries and build a scientific tradition in the field. The entrepreneurial role of US higher education institutions when applying for funding or creating their infrastructural resources has helped shift European academic institutions’ mindset towards nanotechnology consortiums that do not rely on public funding [43].

India and Canada have also shown relevant growth in the cited countries’ dimensions. Despite being outside of the countries with more publications in the field, their research works are significant in the field. Other scholars often cite Indian and Canadian articles. This mirrors the main sources of funding in the field in the last five years, a scale where the National Research Council of Canada stands in one of the highest positions according to the WoS data, reflecting the country’s investment in developing cutting-edge research on biosensors and nanotechnology. 

The most significant conclusions from both countries’ maps indicate the tight relationships between Europe and the USA. The USA’s central role in the centre of the maps is to help bridge partnerships between Europe, Canada, South America, and India, crucial to expanding the biosensors field with innovative collaborative research models. China has been leading the Asian market and trying to join Western institutions in scientific consortiums since the beginning of the century [44]; nonetheless, it requires further effort for closer integration between the Chinese sphere of influence in biosensor production and Western research centres. New synergies between these two different scientific cultures would benefit both parties.

### 4.4. Organisations

After unravelling the partnerships between countries inside the biosensor scientific production field, in this section, the research switches the level of analysis to investigating collaborations between organisations with more extensive publication records in the area. It is expected that this level of analysis reflects the patterns of the previous section. Nonetheless, the inclusion of the organisations allows us to depict which scientific institutions contribute the most to the distribution in the maps displayed above. 

The organisation with the most prolific track record of publishing articles, notes, and reviews in the biosensors field is the Chinese Academy of Sciences (Figure 5). The Chinese Academy of Sciences (CAS) integrates more than 100 research centres, producing most biosensor-related publications. It also comprises three universities that are based on a mixed-model approach of teaching and research and which do not publish as much as the research centres associated with CAS; however, they make a significant contribution to the amount of work published, occupying a significant position in the map (Figure 6).

At the continental level, some of the best-ranked universities are also the main players in the biosensors field; in North America, Harvard Medical School (green; USA) dominates the core of the denominated Western world. In South America, the University of Sao Paulo (red; Brazil) is one of the examples of the rest of the BRICS’s effort to pursue China’s position in this scientific domain. Other universities from Asia are also finding room in this field, such as the University of Tokyo (purple; Japan), Korea University (light blue; South Korea), and Zhejiang University (blue; China), which are also close to the core of the network analysis, despite not achieving the CAS publication numbers. While the yellow/green light cluster aggregates institutions from Turkey and the Middle East, the orange niche has more fundamental representatives in Taiwanese institutions. It brings together organisations from Southeast Asia and Oceania.

There is considerable heterogeneity in the influential organisations that contribute to advancing the state of the art in the biosensors field. Some scientific institution clusters work autonomously, reinforcing the lack of communication between research centres from different geographic locations with diverse ways of advancing science. The map’s dispersed yellow and orange dots might explain this isolation phenomenon. 

The network distribution of the organisations by the number of citations (cf. Appendix A) is more homogeneous than the one based on the number of published articles. The clusters generated are still similar to Figure 5; however, the aggregation of institutions close to the centre is denser when compared with the previous network. 

Given the path that must be taken towards bridging the gap between research culture and idioms and finding new forms of collaboration, globalisation increased scientific partnerships between organisations worldwide. The large agglomerates in the centre of Figure 5 and Figure 6 are examples of this pattern. The synergies emerging from these international scientific consortiums help create knowledge communities that allow mutual research citations and bring scientists together from different areas of study, such as chemistry, healthcare studies, economics, and many others. This phenomenon broadens the spectrum of biosensor production analysis and helps the field’s competencies to become more diverse. 

R&D institutions in different geographical settings think differently. One of the keywords highlighted in the keyword maps is the merging of ideas and investment in polyphasic taxonomy processes, which challenge scientists to find a consensus through knowledge exchange with cultures with diverse ways of making ends meet.

## 5. What Is Next for the Future of Biosensors?

All the possibilities for the biosensors field are still to be explored; new application areas are emerging as the literature focuses more on biosensors as an autonomous area of activity that can adapt itself to the needs of customers and in-need populations. The network maps indicated a trend of scientific organisations from different geographical locations working together towards the progression of research and innovation. The fostering of new technology production methods related to a sustainable transition and green chemistry principles puts European countries such as Germany, the UK, and France at the forefront of the future of biosensors. This expansion is followed by the rise of the BRICS, which might threaten the CAS’s dominance in Asia. The tendency to bring different epistemic communities together through partnerships, the USA–China being the most prominent example, contributes to global knowledge transfer and mitigates possible commercialisation hardships for customers in terms of price. 

The optimism of the biosensors field gravitates around these devices being a critical personal healthcare tool, although the field still needs further exploration. The spectrum of applications is mainly framed by the possibilities of biosensing to engage with the Internet of Things [45,46] and big data through wireless gadgets [47]. The prospect of combining the regular monitoring of health-related levels such as sweat glucose [48], blood test modules [4], or saliva [49] can be fundamental to cancer detection, which is a significant knowledge base in the field today. 

The cyber potential of wearable devices creates room for a new health market. Nevertheless, mixing the development of biosensors with the digitalisation of personal data opens new challenges concerning data management and ethical procedures for health information. An international policy framework for data protection when using self-tracking devices is required for civil society to feel safe using biosensors.

As mentioned by the academic literature, improving performance and design is essential for the widespread adoption and mass production of biosensors. In order to be successful and competitive, a cost-effective multiplex solution must be found that maintains high sensitivity. Additionally, quality control tests in biosensor production should contribute towards reducing storage costs while facilitating biosensors’ introduction to the market [50].

The future of the biosensors field is as important as its present; for that reason, it is crucial to understand the envisioned next steps for scientific partnerships within the biosensors literature. To support the mass commercialisation of the product, strategic decisions must be made to prevent financial and environmental losses so that the biosensors field can grow, supported by a solid foundation of sustainable chemistry methods in an interdisciplinary and international framework.

## 6. Conclusions

Substantial investment at the industrial and scientific levels has pushed biosensing technology forward. The increasing number of publications and patents registered in international databases proves this development. The broad range of biosensor applications, from the chemistry field to the healthcare sector and environmental monitoring, holds potential for a sustainable transition in the future that is expected to be marked by ecological risks and public health hazards. The novelty and diverse possible applications for multiplex biosensors have turned the field into unstable ground where practical use, economic profit, environmental hazards, and social risks are part of a fragile equilibrium where all vertices must be aligned to produce a sustainable and user-friendly product. This product will generate quicker and more reliable results with reduced environmental impacts.

The field’s knowledge base is moving towards scientific methods related to the sustainable transformation of research, which is concerned with developing green chemistry concepts. Currently, the envisioned biosensor applications are predominantly connected to healthcare and biomedicine purposes, specifically diagnosis and tackling new diseases, while improving device characteristics. Performance and design are two features that generate more scientific progress towards turning biosensing devices more accurate, stable, and smaller. 

Within this innovation process, tension has been emerging in the literature related to the delicate balance between scientific achievements through a transformation of the area of studies for a more sustainable field and the reliability of the product to be commercialised. In the overall context of nanotechnology, the complex synthesis procedures of nanoparticles should be gradually improved to ensure that it is possible to bridge the commercial innovation side of the biosensor market with product reliability. 

Moving these advancements in biosensor technology from the lab to the market requires research on how the population would interact with different designs or performance errors. Therefore, international stakeholders should keep promoting efforts to draw up a framework that can ensure a balance between the natural and social sciences in terms of all the factors that gravitate around researching new biotechnology production.

This interdisciplinarity calls for merging diverse research realities that bring new science management approaches and models to biosensor production. The high levels of scientific production in Asia need a more robust network of collaborations to reach a global audience. Meanwhile, the solid connections between research institutions in Europe and the USA might benefit from higher publication levels such as the one achieved by CAS and other Asian institutions. Building synergies for epistemic communities that have regional scopes to become global and improving the intercontinental nature of research consortiums must be two of the main concerns of global funding programmes in the long term. 

Economic, political, and structural factors related to the scientific institutions in which biosensor-related content is produced make this technology an innovation prospect with unknown potential. The literature emphasises the division between academia and enterprises as one of the main bottlenecks in the coming years. This is a debate between science-oriented and profit-oriented innovations. Establishing more synergies between research centres and enterprises in terms of product development and the effective communication of science to civil society are seen as two crucial ways of guaranteeing transformative innovation while maintaining profitable margins [51]. Moreover, in future research, comprehending the scope of this technology is fundamental to tracking the evolution of possible areas of activity and the challenges posed to the future of biosensing research.

## Figures and Tables

**Figure 1 sensors-24-03082-f001:**
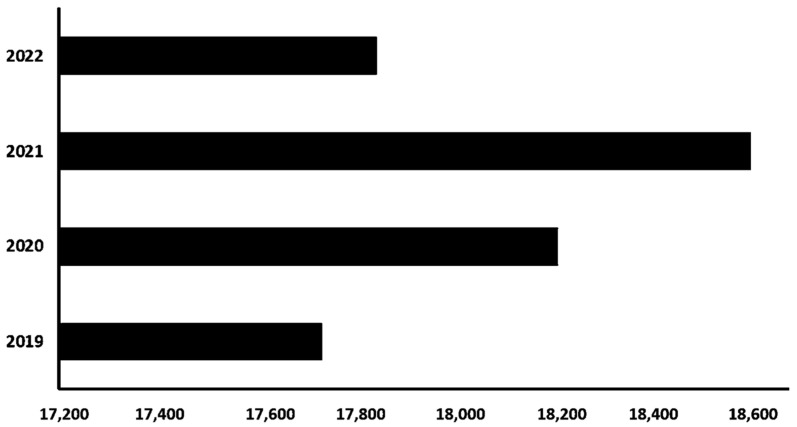
Published literature on biosensor-related topics from 2019 to 2022. (2023 was not considered. The data collected on WoS were available on 10 May 2023). Source: own elaboration based on Web of Science data.

**Figure 2 sensors-24-03082-f002:**
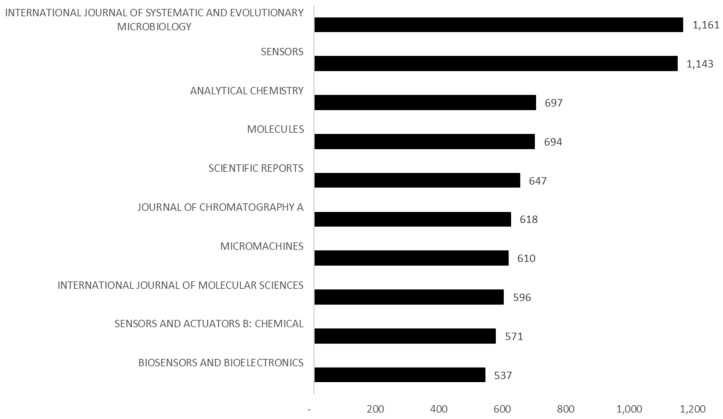
Most frequent sources of biosensor-related articles (2019–2023). Source: own elaboration based on Web of Science data.

**Figure 3 sensors-24-03082-f003:**
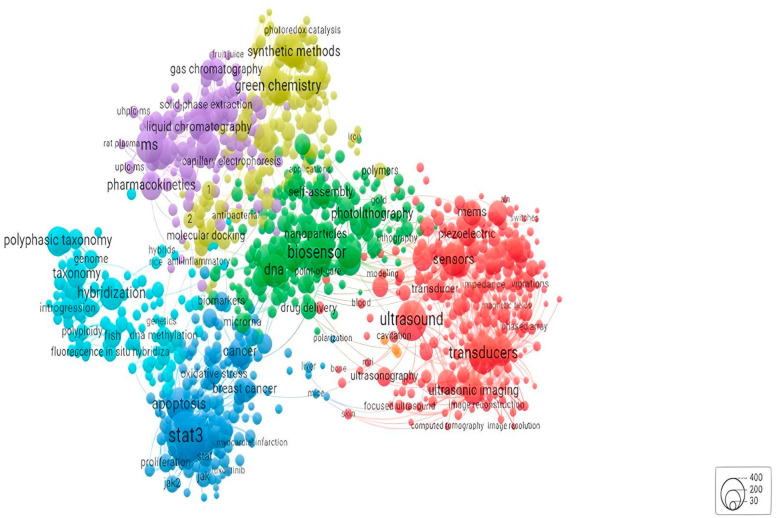
Network distribution by authors’ keywords. Source: own elaboration using VOSviewer based on Web of Science data.

**Figure 4 sensors-24-03082-f004:**
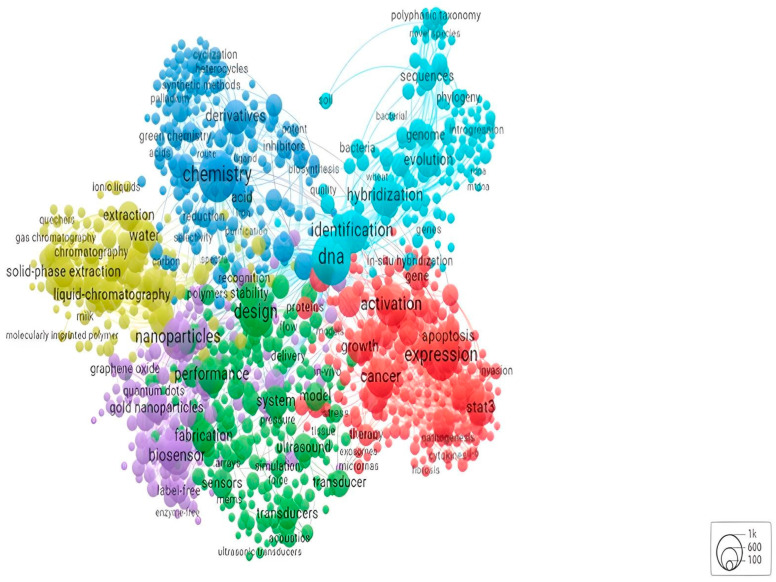
Network distribution by Web of Science keywords. Source: own elaboration using VOSviewer based on Web of Science data.

**Figure 5 sensors-24-03082-f005:**
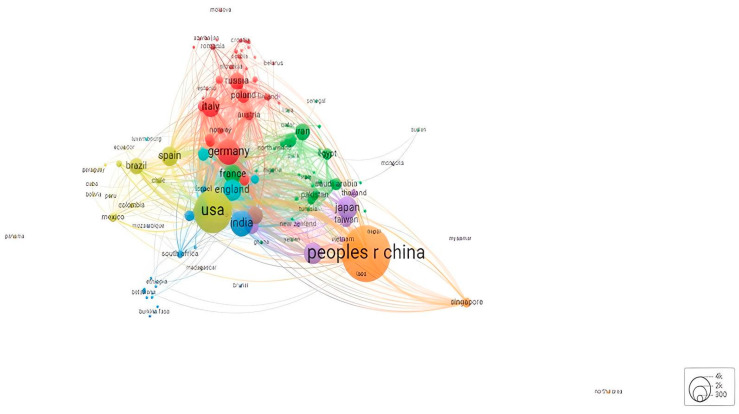
Network distribution of countries by amount of work published. Source: own elaboration using VOSviewer based on Web of Science data.

**Figure 6 sensors-24-03082-f006:**
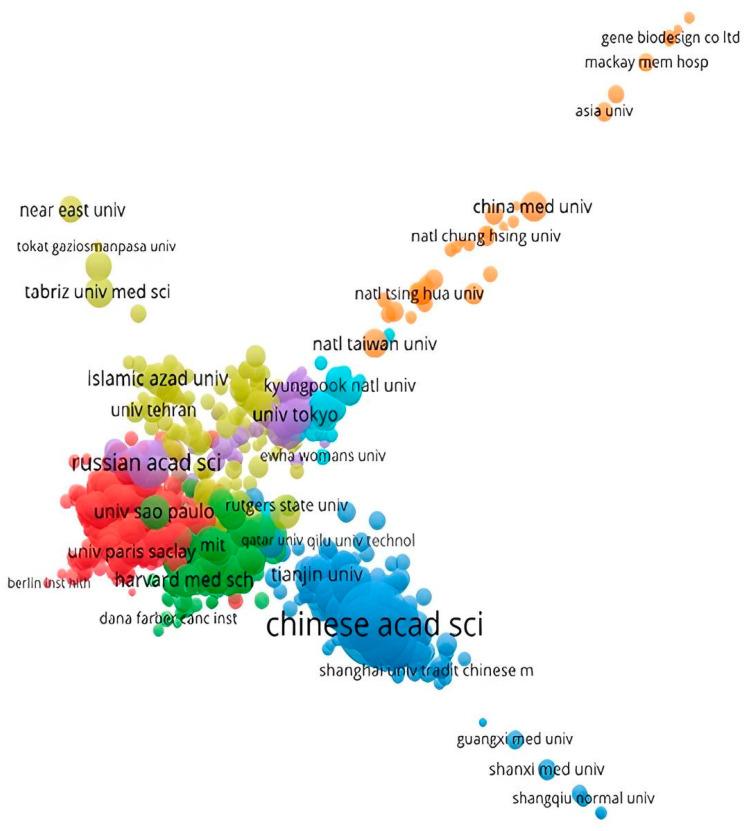
Network distribution of countries by amount of work published. Source: own elaboration using VOSviewer based on Web of Science data.

**Table 1 sensors-24-03082-t001:** Biosensors research areas with more than 1000 publications (2019–2022).

Research Areas ≥ 1000 Publications	Nº of Publications
Chemistry	12,873
Biochemistry and Molecular Biology; Chemistry	3281
Science and Technology—Other Topics	2460
Engineering	2304
Microbiology	2210
Physics	2019
Materials Science	1825
Pharmacology and Pharmacy	1613
Chemistry; Science and Technology—Other Topics; Materials Science; Physics	1409
Optics	1288
Materials Science; Physics	1199
Chemistry; Engineering; Instruments and Instrumentation	1143
Oncology	1057
Total	77,391

Source: own elaboration based on Web of Science data.

**Table 2 sensors-24-03082-t002:** Biosensors Web of Science categories with more than 1000 publications (2019–2022).

Web of Science Categories ≥ 1000 Publications	Nº of Publications
Chemistry, Multidisciplinary	4969
Chemistry, Analytical	4230
Multidisciplinary Sciences	2353
Microbiology	2210
Chemistry, Organic	1944
Biochemical Research Methods; Chemistry, Analytical	1653
Biochemistry and Molecular Biology; Chemistry, Multidisciplinary	1332
Optics	1288
Physics, Applied	1185
Chemistry, Analytical; Engineering, Electrical and Electronic; Instruments and Instrumentation	1143
Oncology	1057
Pharmacology and Pharmacy	1003
Total	77,391

Source: own elaboration based on Web of Science data.

## Data Availability

Data can be found in the database of the Web of Science at https://www.webofscience.com/wos (accessed on 8 March 2022).

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
