# Peer review of "Exploring Biosensors’ Scientific Production and Research Patterns: A Bibliometric Analysis"

_sensors, 2024, doi:10.3390/s24103082_

Round 1
Reviewer 1 Report
Comments and Suggestions for Authors|
This article Tracks the latest advancements in biosensors’ scientific production and reviews the global progress of biosensors. I think it's critical for researchers to understand what's going on in the field. The article can be accepted if the following questions are modified. 1) the key word such as Microfluidic chips, paper-based sensing, integrated biosensor should be included and analyzed. 2) as far as I know, the Talanta journal and ACS sensor journal are also the preferred by those in the field of sensor research. 3)Please unify the format of the references, some with DOI and some without 4)biomarker detection-based biosensor, which include DNA, RNA, protein, exosome etc. should be added and analyzed. 5) As we all know, the innovation point is important in the paper publication. Otherwise, the reliability is important in the commercial product. How to balance these two points. For example, the nanomaterial or nanoparticle-based biosensors which include complicate process of synthesis of nanoparticle or nanomaterial is difficult to grow to be a product. 6) The work of research institutions is innovation in scientific research, and the work of companies is products and profits. How to balance the two and promote scientific research results from shelves to shelves? Can this analysis be added into the article? |
|
|
Author Response
The authors would like to thank the referees for their comments and insights, which raised important issues and added scientific value to our article.
Above, you can find the replies to all the feedback.
Ref 1.
This article Tracks the latest advancements in biosensors’ scientific production and reviews the global progress of biosensors. I think it's critical for researchers to understand what's going on in the field. The article can be accepted if the following questions are modified.
Authors: Thank you!
1) the key word such as Microfluidic chips, paper-based sensing, integrated biosensor should be included and analyzed.
Authors: The selection of keywords was validated collectively, not only by the authors but also by other members of the consortium that are experts in specific aspects of the field. This was clarified in the body of the text. Nevertheless, some of these words were now tested. However, they do not add a very significant number of articles to impact the overall analysis (most of the articles obtained with these keywords were already in the original database).
2) as far as I know, the Talanta journal and ACS sensor journal are also the preferred by those in the field of sensor research.
Authors: We acknowledge the importance of the Talanta journal and ACS Sensor as two of the highest impact factor journals in the field, but based on the amount of work published, they are not within the top 10 of scientific production, even though are extremely influential in the field.
3)Please unify the format of the references, some with DOI and some without
Authors: Done
4)biomarker detection-based biosensor, which include DNA, RNA, protein, exosome etc. should be added and analyzed.
Same as point 1. We also included these as additional keywords.
5) As we all know, the innovation point is important in the paper publication. Otherwise, the reliability is important in the commercial product. How to balance these two points. For example, the nanomaterial or nanoparticle-based biosensors which include complicate process of synthesis of nanoparticle or nanomaterial is difficult to grow to be a product.
Authors: We added a paragraph in the conclusion section to address this concern.
6) The work of research institutions is innovation in scientific research, and the work of companies is products and profits. How to balance the two and promote scientific research results from shelves to shelves? Can this analysis be added into the article?
Authors: We also added this to stimulate further reflection in the conclusion.
Reviewer 2 Report
Comments and Suggestions for Authors
This work is a nice report of a bibliometric analysis of biosensors. The overall exposition is clear and well documented, and the highlights were presented in the appropriate way. However, I would like to suggest a few of minor corrections as follows:
-The body content is little deviated from title, a modification in content or title is suggested.
- Figure 2 should be completed with the actual impact factor of each journal to better appreciation of data.
- In keyword selection: Photolithograph*, Transducer*; Multiplex Biosensors*; Wafer*; Microchip*; DNA Hybrid*; Synthetic Chemistry*; Silicon Biosensor*; Microelectromechanical Sensor*. And all the study derivated from those words, I´m really afraid if important information come from i.e electrochemical biosensors (which are very relevant in this área), and monoplex biosensors (which so far are the most optimized) is lost from analysis.
Author Response
Ref 2.
This work is a nice report of a bibliometric analysis of biosensors. The overall exposition is clear and well documented, and the highlights were presented in the appropriate way.
Authors: Thank you!
However, I would like to suggest a few of minor corrections as follows:
-The body content is little deviated from title, a modification in content or title is suggested.
Authors: We changed the title, to a more comprehensive option that describes the work done in this paper.
- Figure 2 should be completed with the actual impact factor of each journal to better appreciation of data.
Authors: A new table was added with this additional information in the appendix section.
- In keyword selection: Photolithograph*, Transducer*; Multiplex Biosensors*; Wafer*; Microchip*; DNA Hybrid*; Synthetic Chemistry*; Silicon Biosensor*; Microelectromechanical Sensor*. And all the study derivated from those words, I´m really afraid if important information come from i.e electrochemical biosensors (which are very relevant in this área), and monoplex biosensors (which so far are the most optimized) is lost from analysis.
Authors: The selection of keywords was validated collectively, not only by the authors but also by other members of the consortium that are experts in specific aspects of the field. This was better clarified in the body of the text in the current version. Nevertheless, some of these words were now tested. However, they do not add a very significant number of new articles to impact the overall analysis (most of the articles obtained with these keywords were already in the original database).
Kind regards,
The authors